# Effect of Organizational Evolution on the Stress Corrosion Cracking of the Cr-Co-Ni-Mo Series of Ultra-High Strength Stainless Steel

**DOI:** 10.3390/ma15020497

**Published:** 2022-01-10

**Authors:** Shuai Tian, Zhenbao Liu, Renli Fu, Chaofang Dong, Xiaohui Wang

**Affiliations:** 1College of Materials Science and Technology, Nanjing University of Aeronautics and Astronautics, Nanjing 210016, China; shuai_tian2076@163.com; 2Institute for Special Steel Institute, Central Iron and Steel Research Institute, Beijing 100081, China; wangxiaohui@nercast.com; 3Corrosion and Protection Center, University of Science and Technology Beijing, Beijing 100083, China; cfdong@ustb.edu.cn

**Keywords:** ultra-high strength stainless steel, SCC, laves phase, austenite

## Abstract

Different microstructures were obtained under various thermal conditions by adjusting the heat treatment parameters of the Cr-Co-Ni-Mo series of ultra-high strength stainless steel. The effect of organizational evolution on the stress corrosion cracking (SCC) of the Cr-Co-Ni-Mo series of ultra-high strength stainless steel was investigated using potentiodynamic polarization curves, electrochemical impedance spectroscopy (EIS), transmission electron microscopy (TEM), scanning electron microscopy (SEM) and other test methods in combination with slow strain rate tensile tests (SSRTs). The results show that the Mo- and Cr-rich clusters and precipitation of the Laves phase reduce the corrosion resistance, while increasing the austenite content can improve the corrosion resistance. The Cr-Co-Ni-Mo series of ultra-high strength stainless steel has a high SCC resistance after quenching at 1080 °C and undergoing deep cooling (DC) treatment at −73 °C. With increasing holding time, the strength of the underaged and peak-aged specimens increases, but the passivation and SCC resistance decreases. At the overaged temperature, the specimen has good SCC resistance after a short holding time, which is attributed to its higher austenite content and lower dislocation density. As a stable hydrogen trap in steel, austenite effectively improves the SCC resistance of steel. However, under the coupled action of hydrogen and stress, martensitic transformation occurs due to the decrease in the lamination energy of austenite, and the weak martensitic interface becomes the preferred location for crack initiation and propagation.

## 1. Introduction

Ultra-high strength stainless steel is widely used in aviation marine and other fields such as aircraft landing gear, wing girder and bolts due to its high strength, high toughness and good corrosion resistance [1,2,3]. The excellent mechanical properties of ultra-high strength stainless steel is attributed to martensitic transformation strengthening [4] and precipitation strengthening [5]. To obtain a higher strength, researchers have carried out considerable research, especially on the precipitation strengthening behaviour of ultra-high strength stainless steel with different strength grades. For example, Habibi Bajguirani [6] et al. investigated the precipitation strengthening behaviour of 15-5PH and showed that the formation of Cu-rich precipitates had a significant precipitation strengthening effect and that the tensile strength could reach 1240 MPa, while the overaging treatment coarsened the Cu-rich precipitates, which in turn led to a decrease in the strength of the steel. Researchers developed PH13-8Mo with higher strength by reducing the Cr content and increasing the Ni content based on the 15-5PH alloy ratio. The strength of PH13-8Mo is enhanced by precipitation strengthening of the β-NiAl precipitates with a B2 superlattice structure [7,8]. However, with increasing strength, a new problem, namely stress corrosion cracking (SCC), occurs and the SCC susceptibility increases with increasing strength [9,10,11]. Therefore, it is of great practical significance to study the SCC of ultra-high strength stainless steel.

When ultra-high strength stainless steel is applied as a strained component in an environment containing erosive ions (such as Cl^−^), SCC mainly originates in pitting. The nucleation of pitting is influenced by various factors, among which inclusions, precipitates, grain boundaries, passivation film rupture, and plastic deformation are the main factors that cause pitting nucleation in ultra-high strength stainless steel [12,13,14]. As the main strengthening phase in steel, it is very important to study the effect of precipitates on pitting nucleation. For this reason, the influence of precipitates on pitting nucleation has been studied extensively [15,16,17]. It has been found that the precipitates of Cr-rich carbides at grain boundaries leads to Cr-poor zones around them, which in turn trigger pitting nucleation. For example, Luo et al. investigated the effect of holding time on the organization and electrochemical behaviour of precipitation-hardened stainless steel (15-5 PH) using various techniques, such as a three-dimensional atom probe (3DAP), and found that (Cu, Nb)-rich MC nanoparticles began to precipitate with increasing holding time, while Cr-poor zones appeared near the (Cu, Nb)-rich MC nanoparticles, which led to nucleation by pitting [18]. Chen et al. [19] found that the strain energy at the interface between the G-phase and ferrite matrix was higher than that of the intracrystalline atoms and was prone to react with Cl^−^ in solution to form pitting corrosion. In addition, scholars [20,21,22] found that σ and χ precipitates had a significant effect on the pitting of stainless steels.

The SCC crack initiation process of ultra-high strength stainless steel is controlled by pitting, while the crack propagation stage is mainly controlled by anodic dissolution (AD) and hydrogen embrittlement (HE) mechanisms [23,24]. Among them, the AD mechanism contains the slip dissolution mechanism [25], the oxide film cracking mechanism [26], and the selective dissolution mechanism [27]. The HE mechanisms include hydrogen-enhanced decohesion (HEDE) [28], hydrogen-enhanced localized plasticity (HELP) [29], and adsorption-induced dislocation emission (AIDE) [30]. There is considerable evidence [31,32] that H plays a key role in SCC. For example, Beaver and Harle noted that SCC was caused by AD and propagated by the HE mechanism [33]. In fact, the H generated by metal cation hydrolysis at the crack tip and cathodic hydrogen evolution would increase the sensitivity of HE during the SCC process of ultra-high strength stainless steel. Therefore, the occurrence of HE could be effectively reduced by introducing benign H trapping sites to capture H or slow down the migration of H to the metal lattice. The existence of hydrogen traps was first proposed by Darken and Smith in 1949. Currently, the known hydrogen traps in steel include dislocations [34], grain boundaries [35], inclusions [36], precipitates [37], austenite [38], etc. Li et al. [39] found that the resistance to HE of the specimens correlated well with the content of austenite. He believed that it was feasible to improve the HE resistance by increasing the content of austenite. Tsay et al. [40] also obtained similar results. However, Fan et al. [41] argued that the beneficial effect of austenite should not be overestimated since cracking along the tempered martensite/newly generated martensite boundary occurred after martensitic transformation due to the redistribution of hydrogen. Therefore, the effect of austenite on the HE of steel has to be further confirmed.

In summary, the organizational evolution of ultra-high strength stainless steel, such as the formation of precipitates, content of austenite and dislocation density, has an important influence on SCC [42]. The purpose of this study is to obtain microstructures under different thermal conditions by adjusting the heat treatment process and to study the effect of microstructural evolution on the SCC of the Cr-Co-Ni-Mo series of ultra-high strength stainless steel in nearly neutral 3.5 wt.% NaCl solution. The mechanism of SCC crack nucleation and crack propagation is also analysed and discussed.

## 2. Experimental Methods

### 2.1. Preparation of Specimens and Experimental Methods

Vacuum induction and consumable-electrode vacuum melting were used to obtain Cr-Co-Ni-Mo ultra-high strength stainless steel. After melting, the ingots were heated at 1160 °C and forged into *ϕ*450 mm round bars. A 20 mm × 20 mm × 20 mm sample was cut from the bar for measuring chemical composition, and the result is shown in Table 1. All specimens were cut from the round bars by wire-electrode cutting. The specimens were solution-treated at 1080 °C for 60 min, quenched in oil at room temperature and then cryogenically treated (deep cooling (DC)) at −73 °C for 8 h. Finally, the specimens were aged at 480 °C, 540 °C and 600 °C for 0.5 h, 4 h and 80 h, respectively, and these thermal conditions were designated 480A-0.5/4/80, 540A-0.5/4/80, and 600A-0.5/4/80, respectively.

The SSRT specimens with a diameter of 3 mm and a gauge length of 23 mm were prepared according to the requirements of GB/T15970.7-2000 [43], and the specimens were first sequentially sanded with #1000, #1500 and #2000 sandpaper and then cleaned in alcohol solution. The SSRTs were performed on tensile specimens in air and 3.5 wt.% NaCl solution at a constant strain rate of 10^−6^ s^−1^ using a WDML-300 kN testing machine.

The relative plasticity loss was used to evaluate the SCC susceptibility of the specimens in 3.5 wt.% NaCl solution. The relative loss of postextension and section shrinkage of the specimens were calculated using Equations (1) and (2), respectively.
(1)$$\varphi_{loss}=\frac{\varphi_{A}-\varphi_{S}}{\varphi_{A}}\times 100\%$$
(2)$$\delta_{loss}=\frac{\delta_{A}-\delta_{s}}{\delta_{A}}\times 100\%$$
where *φ**_loss_* and *δ**_loss_*are the relative reduction in the area and elongation to fracture, respectively, *φ**_A_*and *φ**_S_* are the area reduction of the specimens in air and 3.5 wt.% NaCl solution, respectively, *δ_A_* represents the elongation to fracture of the specimens in air, and *δ_S_* stands for the elongation to fracture of the specimens in 3.5 wt.% NaCl solution.

### 2.2. Electrochemical Analyses

Potentiodynamic polarization and electrochemical impedance spectroscopy (EIS) were performed with a Zennium pro electrochemical workstation. A three-electrode system, in which a 10 mm × 10 mm × 5 mm specimen was the working electrode, a platinum sheet was the auxiliary electrode, and a saturated calomel electrode (SCE) was the reference electrode, was used for electrochemical measurements in 3.5 wt.% NaCl solution. First, a constant potential polarization of −1 V vs. SCE was applied for 800 s to remove the oxide film. Then, an AC amplitude of 10 mV was applied to the open circuit potential (OCP), and EIS was performed in the frequency range of 100 kHz to 10 mHz. ZsimpWin software (1.0.0.0) was used to merge impedance data and establish an equivalent circuit diagram. Finally, the potentiodynamic polarization curves were measured by a scanning rate of 50 mV/s, and the scanning potential range was −1.0–0.5 V.

### 2.3. Microstructure Characterization and Fracture Analysis

Specimens for microstructural observations were ground and polished following conventional metallographic standard procedures and then etched with Fry’s reagent (1 g CuCl_2_ + 50 mL HCl + 25 mL HNO_3_ + 150 mL H_2_O). The microstructures of the specimens were observed using a Zeiss-40 MAT optical microscope (OM, Carl Zeiss AG, Oberkochen, Germany). The volume fraction of austenite, dislocation density and precipitates extracted by electrolysis were determined by a Bruker D8 Advance X-ray diffractometer (Bruker, Karlsruhe, Germany) with a Co-Kα radiation source operated at a voltage of 35 kV and a current of 40 mA. The scanning rate and step size were 5° min^−1^ and 0.02°, respectively. The volume fraction of austenite and dislocation density were processed using a modified Williamson-Hall method [44].

The morphology and distribution of austenite in the sample were characterized by electron back-scattered diffraction (EBSD, FEI, Hillsboro, OR, USA). The specimen size was 8 mm × 5 mm × 1 mm. The surface of the specimens was slightly polished with a velvet polishing cloth and 2.5 µm polishing paste until the surface of the specimens was bright and unpolluted. Then, the sample was electropolished to remove the surface stress layer and tested by EBSD. Transmission electron microscopy (TEM, FEI Talos F200X, FEI, Hillsboro, OR, USA) was used to observe the fine tissue. The TEM specimens were prepared by mechanically grinding thin wafers to a thickness of 40 μm. Then, the wafers were jet polished in a 10 vol.% HClO_4_ methanol electrolyte at −20 °C with a constant current of 50 mA. The fractured specimens were cleaned in an ultrasonic cleaning machine, and the fracture morphology of the specimens was observed by manual emission scanning electron microscopy (SEM, FEI Quanta 650, FEI, Hillsboro, OR, USA) after cleaning.

## 3. Results

### 3.1. Microstructure

An optical micrograph of the prior austenite grain boundaries (PAGBs) of DC and the microstructure of the specimens under different thermal conditions are shown in Figure 1. The average grain size of DC measured by Nano Measure image analysis software is approximately 96.28 μm (Figure 1a), and the metallographic structure of the specimens is typical lath martensite (Figure 1b). As shown in Figure 1c–k, the metallographic organisations of the specimens were all typical lath martensite, and the martensite structure became vague upon increasing the aging temperature and extending the aging time.

Figure 2 shows the XRD patterns of specimens under various thermal conditions. Figure 2a shows that the XRD patterns of the specimens have obvious martensite diffraction peaks and austenite diffraction peaks, but the diffraction peak intensities are different. For example, (111)_γ_ shows stronger intensity in the specimens aged at a higher temperature or held for a longer holding time, which indicates that the volume fraction of austenite increases. Figure 2b shows that the volume fraction of austenite increases with increasing ageing temperature and holding time. The austenite content of DC is approximately 3.52%. When the holding time is extended from 0.5 h to 80 h, the austenite content of the specimens aged at 480 °C increases from 4.35% to 7.72%, and the austenite content of the specimens aged at 540 °C and 600 °C increases from 6.15% to 12.82% and 8.40% to 23.56%, respectively.

Figure 2c shows the dislocation density of specimens under different thermal conditions. The dislocation density of the specimens gradually decreases with increasing ageing temperature and holding time. DC has the highest dislocation density of approximately 6.1 × 10^11^/cm^2^. With the increase in holding time from 0.5 h to 80 h, the dislocation density decreases as follows: 480 °C-aged specimens from 5.3 × 10^11^/cm^2^ to 4.6 × 1011/cm^2^, 540 °C-aged specimens from 5.1 × 10^11^/cm^2^ to 2.5 × 10^11^/cm^2^ and 600 °C-aged specimens from 1.8 × 10^11^/cm^2^ to 2.2 × 10^10^/cm^2^.

Figure 3a,b show the TEM micrographs of DC. The microstructure of DC is composed of lath martensite with a high dislocation density, and the width of the martensite lath is approximately 100~400 nm. Figure 3c–h show the TEM morphology after ageing at 480 °C for various holding times. It can be seen from Figure 3c,d that the TEM microstructure after holding for 0.5 h still shows lath martensite with a high dislocation density, and the dislocation density decreases with the increasing holding time, which is related to the recovery of dislocations with the holding time. No obvious precipitates are found when the holding time was 4 h. The clusters form obviously at 80 h and grow gradually with increasing holding time, as shown in Figure 3f,h.

Figure 4 shows the TEM micrographs of the specimens aged at 540 °C for different holding times. As shown in Figure 4c,d, when the holding time is 4 h, a large number of small and uniform precipitates begin to precipitate. After increasing the holding time to 80 h, the precipitates grow. From the diffraction spot calibration results, it can be seen that the precipitates are the Laves phase of the hexagonal system, as shown in Figure 4e,f. Figure 5 shows the TEM micrographs of the samples aged at 600 °C for various holding times. Compared with the precipitates at lower ageing temperatures, the precipitates precipitate after ageing at 600 °C for 0.5 h, with a size of approximately 30 nm (Figure 5a,b). A large number of precipitates with a size of approximately 40 nm precipitate in the specimens with a holding time of 4 h, as shown in Figure 5c,d. When the holding time is increased to 80 h, the precipitates are significantly coarsened, as shown in Figure 5e. From the diffraction spot calibration results of the precipitates in Figure 5f, it can be seen that the precipitates in 600A-80 are still the Laves phase with a size of approximately 50 nm.

### 3.2. Results of the Electrochemical Tests

Figure 6 shows the potentiodynamic polarization curve of the specimens in 3.5 wt.% NaCl solution. Figure 6a shows that DC is passivated in 3.5 wt.% NaCl solution, which is caused by the oxide or hydroxide film of Fe, Cr, Ni, Mo and other elements on the surface of the specimens. When the potential is higher than 0.193 V_SCE_, a large increase in the current density indicates that the rupture of the local passivation film causes anodic dissolution and pitting. As shown in Figure 6b,c, the electrochemical behaviour of the specimen changes significantly with the increasing holding time. It can be seen from the curve that the samples with holding times of 0.5 h and 4 h are passivated, while the specimens with holding times of 80 h are in a state of activation and dissolution, indicating that the corrosion resistance of the matrix is reduced by the long ageing treatment. Figure 6d shows that there are passivation intervals for specimens with different holding times. However, with the increasing holding time, the passivation interval decreases, which indicates that the corrosion resistance of each sample decreases with the increasing holding time. The potentiodynamic polarization curve was fitted using Thales XT software. The fitting results of the potentiodynamic polarization curve of the specimens are shown in Table 2. At the same ageing temperature, E_coor_ and E_pit_ decrease with increasing holding time, and I_corr_ and I_P_ increase with increasing holding time, indicating that a long ageing treatment reduces the corrosion resistance of the specimens.

Figure 7 shows that the Nyquist diagrams of the specimens in 3.5 wt.% NaCl solution. As shown in Figure 7a, the capacitance arc radius of DC is larger than that of the aged specimens, which indicates that the ageing treatment reduces the corrosion resistance of the specimens. As shown in Figure 7b–d, at the same ageing temperature, the capacitive reactance arc decreases with the increasing holding time. However, the capacitance arc radius of 600A-80 is significantly greater than that of 480A-80 and 540A-80. The EIS data are fitted according to the equivalent circuit shown in Figure 8, and the fitting results are shown in Table 3. In the above circuit, R_S_ is the solution resistance, R_1_ and R_2_ are the passive film resistance and charge transfer resistance, respectively, CPE1 and CPE2 are the passive film capacitance and double-layer capacitance, respectively, and n is the dispersion coefficient. Generally, the polarization resistance R_P_ (the sum of R_1_ and R_2_) can be used to evaluate the corrosion resistance of specimens. Table 3 shows that with the increasing holding time, the polarization resistance and corrosion resistance of the samples decrease, which is consistent with the test results of the potentiodynamic polarization curve.

### 3.3. Tensile Properties

The engineering stress-displacement curves of the specimens in air and 3.5 wt.% NaCl solution are shown in Figure 9. Table 4 shows the detailed results of the SSRTs. Table 4 shows that 540A-4 has a high strength and a good plasticity. According to the comprehensive mechanical properties, the peak ageing condition is 540 °C for 4 h. Therefore, 480 °C and 600 °C are defined as the underaged and overaged temperatures, respectively. As shown in Figure 9a, the engineering stress-displacement curves of DC in air and 3.5 wt.% NaCl solution almost coincide, indicating that DC has no SCC susceptibility. It can be seen from Figure 9b,c that there is a significant difference in the engineering stress-displacement curve of the sample in air and 3.5 wt.% NaCl solution. Compared with the specimens in air, the tensile strength and plasticity of the sample in 3.5 wt.% NaCl solution decreases significantly with the increasing holding time. This shows that at the underaged and overaged temperatures, the SCC susceptibility of the samples increases with the increasing holding time. As shown in Figure 9d, 600A-0.5 does not show obvious SCC susceptibility, but the SCC susceptibility first increases and then decreases with the increasing holding time. This is related to the precipitates, austenite content and dislocation density in steel, which will be discussed in Section 4.

### 3.4. Fractography

Figure 10 exhibits the fracture morphology of DC. As shown in Figure 10a,d, the fracture of the DC in air and 3.5 wt.% NaCl solution is composed of a central fibre zone and an external shear lip zone. From the local enlarged view of the fibre zone, it can be seen that the fibre zone contains a large number of deep dimples with high tear edges, as shown in Figure 10b,e. The shear lip zone is composed of many small and shallow dimples, as shown in Figure 10c,f.

Figure 11 shows the SEM fractographs of the sample aged at 480 °C after SSRTs in air and 3.5 wt.% NaCl solution. Figure 11a,c,e show that the fracture of the sample in air presents ductile fracture characteristics with obvious necking phenomena, and the fracture is composed of a central fibre zone and an external shear lip zone. When in 3.5 wt.% NaCl solution, the fracture morphology changes significantly. This change is mainly manifested where the crack source of the sample in air appears in the fibre zone, and this fracture is caused by plastic deformation. In contrast, the crack source appears on the surface of the sample in 3.5 wt.% NaCl solution, and this fracture is caused by pitting corrosion. In addition, the fracture morphology of 480A-0.5 and 480A-4 in 3.5 wt.% NaCl solution consists of quasi-cleavage (QC) and intergranular (IG) fracture in the crack source zone and crack propagation zone and the shallow dimple morphology of the transient fracture zone, as shown in Figure 11b,d. Compared with 480A-0.5, the area of QC + IG fracture of 480A-4 significantly increases, and the necking phenomenon decreases, which indicates that the SCC sensitivity increases. The 480A-80 fracture is a typical brittle fracture with a flat morphology. The fracture is mainly composed of QC in the central zone and a small amount of shallow dimples at the edge, as shown in Figure 11f.

The fracture morphology of samples aged at 540 °C for different holding times in air and 3.5 wt.% NaCl solution is basically consistent with that of specimens aged at 480 °C, as shown in Figure 12a–f. The difference is that the areas of the fibre zone and shallow dimple zone of 540A-80 in air are larger than 480A-80, indicating that the plasticity of 540A-80 clearly decreases. In addition, at the same holding time, compared with 480 °C aged samples, the QC ratio of crack source zone and propagation zone of fracture aged at 540 °C in 3.5 wt.% NaCl solution decreases and the IG ratio increases.

As shown in Figure 13a,b, there is no obvious difference in the fracture morphology of 600A-0.5 in air and 3.5 wt.% NaCl solution, and the fracture is caused by plastic deformation. As shown in Figure 13c, 600A-4 in air does not have obvious necking, and the plasticity decreases significantly compared with 480A-4 and 500A-4. This is attributed to the coarsening of the Laves phase, the increase in the resistance to dislocation movement and the decrease in plasticity. A similar phenomenon also occurs in the fracture morphology of 600A-80 in air, as shown in Figure 13e. Interestingly, the proportion of the shallow dimple zone of 600A-80 in 3.5 wt.% NaCl solution is significantly higher than that of 480A-80 and 540A-80, as shown in Figure 13f. Table 4 shows that the SCC susceptibility of 600A-80 is significantly lower than that of 480A-80 and 540A-80, indicating that prolonging the holding time at the overaged temperature can decrease SCC susceptibility.

Figure 14 exhibits the SEM morphology of the brittle fracture zone, plastic fracture zone and corresponding interface zone of different effective specimens in 3.5 wt.% NaCl solution. The fracture of the 480A-0.5 brittle fracture zone is mainly IG and a small amount of QC fracture, while the fracture of the plastic fracture zone is composed of a large number of shallow dimples, as shown in Figure 14a–c. It is worth noting that micropores and tear ridges are observed on the fracture surface of the IG fracture, which is a typical feature of the HE of high strength steel [45,46]. The fracture of the 480A-80 brittle fracture zone is mainly a QC fracture, and the fracture of the plastic fracture zone is composed of a large number of shallow dimples, as shown in Figure 14d–f. As shown in Figure 14g–i, the fracture morphology of 540A-0.5 is similar to that of 480A-0.5, except that the proportion of QC fractures in the brittle zone increases. With an increase in holding time, the fracture morphology of the sample clearly changes, and the brittle zone can be divided into the IG and QC zones. Micropores and tear ridges are observed on the surface of the IG fracture, and secondary cracks are observed on the surface of the QC fracture, as shown in Figure 14j–l. Table 4 shows that 600 A-0.5 has low SCC sensitivity, which is verified in the fracture morphology diagram; that is, the fracture of 600 A-0.5 is composed of a deep dimple morphology in the fibre zone and a shallow dimple morphology at the edge, as shown in Figure 14m–o. The crack source area of 600A-80 shows obvious HE characteristics. As shown in Figure 14q, a large number of secondary cracks are found in the PAGB, which mainly propagate along the martensitic laths. Fan et al. [41] reported that this is because after the martensitic transformation of thin-film austenite in the lath boundary, cracking along the tempered martensite/newly formed martensite boundary will occur due to the redistribution of hydrogen.

## 4. Discussion

### 4.1. Effect of Ageing Treatment on Corrosion Resistance

It can be seen from the potentiodynamic polarization curve in Figure 6 and the Nyquist spectrum in Figure 7 that the ageing treatment has a significant effect on the corrosion resistance of the specimens. The corrosion resistance of DC is obviously better than that of aged specimens. Specifically, the E_corr_, E_pit_ and capacitance arc radius of DC are higher than those of the aged specimens, indicating that ageing treatment reduces the corrosion resistance of the matrix. In addition, at the same ageing temperature, the corrosion resistance of the specimens decreases with the increasing holding time. At the same holding time, the corrosion resistance of the sample decreases first and then increases. Therefore, the change in microstructure during ageing treatment has a significant effect on the corrosion resistance.

Table 5 exhibits the quantitative analysis results of the precipitates. The clusters in 480A-4 are Mo- and Cr-enriched clusters and the precipitates in 540A-4 and 600A-4 are Laves phases rich in Mo- and Cr; notably, the mass fraction of precipitates increases rapidly with the increasing ageing temperature. Figure 15 shows the distribution of Cr, Mo and Ni in 480A-4, 540A-4 and 600A-4 in the matrix. As shown in Figure 15b,c,f,g,j,k, with increasing ageing temperature, the enrichment degree of Mo- and Cr increases, and obvious Mo- and Cr-depleted areas appear around the Mo- and Cr-enriched areas. It is well known that Mo is an effective element to improve the pitting resistance of stainless steel in Cl^−^ solution [47]. Mo can improve the pitting corrosion resistance of stainless steel in the following ways. First, as a passive film-forming element, Mo easily combines with O in solution to form MoO_2_. MoO_2_ can improve the stability of the passive film on the surface of stainless steel, thereby improving the pitting corrosion resistance of stainless steel [48]. Second, MoO_4_^2^^−^ combines with Fe^2+^ to form FeMoO_4_, which is adsorbed around the pits and inhibits pit expansion [49]. Finally, at the bottom of the pit, the accumulation of Mo^6+^ inhibits the growth of pits [50]. Cr is an important component in stainless steel, and the formation of a dense Cr oxide or hydroxide rust layer on the surface of stainless steel will isolate stainless steel from the corrosive environment, greatly decreasing the corrosion rate [51]. Therefore, during the ageing treatment, the precipitation and growth of the Mo- and Cr-enriched clusters and Laves phase lead to the formation of Mo- and Cr-depleted regions around the clusters, which is the main reason for the reduction in corrosion resistance.

It is worth noting that a thin-film Ni-enriched region is formed near the Cr-depleted region in 540A-4 and 600A-4, as shown in Figure 15h,l. It can be seen from the TEM characterization that the thin-film Ni-enriched region in 600A-4 contains reverted austenite, and the morphology and diffraction spot calibration results are shown in Figure 16a–c. Song et al. confirmed that the reverted austenite was enriched by austenite stabilized with Ni, which tended to nucleate in the Cr-depleted region [52]. Previous studies have shown that Ni-enriched reverted austenite has excellent corrosion resistance, and the formation of reverted austenite may slow the decrease in corrosion resistance caused by the formation of a Cr-depleted zone [44]. Table 2 and Table 3 list the electrochemical fitting parameters. The corrosion resistance of 600A-4 is better than that of 480A-4 and 540A-4 because the content of reverted austenite in 600A-4 is 73.64 % and 38.73 % higher than that of 480A-4 and 540A-4, respectively (Figure 2b). In addition, as shown in Figure 2b, 600A-80 has the highest content of reverted austenite, approximately 195.14 % higher than 600A-4. Under the combined action of Laves phase growth and reverted austenite formation, the corrosion resistance of 600A-80 is only slightly lower than 600A-4, but is significantly better than 480A-80 and 540A-80, as shown in Table 2 and Table 3. Therefore, it can be concluded that with an increasing ageing temperature and holding time, the precipitation and growth of the Mo- and Cr-enriched clusters and Laves phase lead to a decrease in corrosion resistance, while the formation of reverted austenite slows the decrease in corrosion resistance.

### 4.2. Effect of Ageing Treatment on SCC

As shown in Table 4, DC has no SCC sensitivity. At the underaged and peak-aged temperatures, the SCC sensitivity increases with increasing holding time. At the overaged temperature, the SCC sensitivity first increases and then decreases with increasing holding time. This is closely related to the increase in the amount of the Laves phase (Figure 5), the increase in reverted austenite content (Figure 2b) and the decrease in dislocation density (Figure 2c) during the ageing treatment. DC does not have SCC sensitivity, mainly due to its uniform structure and its good corrosion resistance. As shown in Table 2 and Table 3, the E_pit_ and polarization resistance of DC are significantly higher than those of aged specimens. The dense passive film on the surface of DC effectively prevents Cl^−^ corrosion [53]. Therefore, DC does not have SCC sensitivity. The increase in the SCC sensitivity of the underaged and peak-aged specimens with increasing holding time is caused for two reasons. First, with the increasing holding time, the precipitation and growth of the Mo- and Cr-enriched clusters and Laves phase with a high number density are observed in the martensitic matrix, resulting in a significant decrease in the corrosion resistance of the specimens. In particular, 480A-80 and 540A-80 are always in the activated dissolution state, as shown in Figure 6b,c, so that the content of H produced by the corrosion reaction increases [54]. H accumulates near the crack tip under the action of stress. When the stress intensity and H concentration at the crack tip reach a certain critical value, HE easily occurs. On the other hand, with the increasing holding time, the strength of the underaged and peak-aged specimens increases, resulting in a decrease in the critical H concentration during fracture [55]. Therefore, the decrease in corrosion resistance and the increase in strength lead to an increase in SCC sensitivity.

The low SCC sensitivity of 600A-0.5 is mainly due to its high austenite content and its low dislocation density in addition to good corrosion resistance. The gap position in austenite is larger than that in martensite, which increases the solubility of H and reduces the diffusion rate [56,57], thus inhibiting the enrichment of H at the crack tip. Dislocation as a reversible hydrogen trap in steel has a certain effect on the transfer of H in the metal lattice. Dislocation-enhanced H transport can lead to greater H penetration in the metal at rates faster than typical diffusion processes [58]. Especially at slow strain rates, dislocations are saturated by H and transport H to the embrittlement region [34], thus increasing SCC sensitivity. Therefore, a high austenite content and low dislocation density can improve the SCC resistance of the specimens. Although, 600A-4 has a higher austenite content and lower dislocation density than 600A-0.5, the corrosion resistance of 600A-4 is lower than that of 600A-0.5, and the precipitation of a large Laves phase in 600A-4 easily causes a stress concentration area. Driven by stress, H is more likely to transfer to the stress concentration area and realize local embrittlement. Therefore, the SCC sensitivity of 600A-4 is higher than that of 600A-0.5.

The SCC sensitivity of 600A-80 is lower, mainly due to having the highest austenite content, as shown in Figure 2b. Figure 17 is the EBSD phase diagram corresponding to specimens under various thermal conditions. Figure 17 shows that there are three types of austenite structures in specimens under various thermal conditions, namely, large blocky, smaller strip-like, and thin-film austenite, and the austenite content increases with increasing ageing temperature and holding time. As shown in Figure 17f, a large amount of austenite in 600A-80 is dispersed in the martensitic matrix. Austenite plays two roles in the SCC of steel: on the one hand, Ni-enriched thin-film austenite is in the thermodynamic steady state and can be used as a ductile phase to passivate the crack tip and improve the crack resistance of the specimen [52]; on the other hand, austenite, as an irreversible hydrogen trap (pinning energy = 55 kJ/mol), can slow the enrichment of the crack tip and reduce the occurrence of HE [59]. However, as austenite is the main H capture point in steel, H can accumulate in austenite. Under the action of H and stress, the stacking fault energy of austenite decreases, martensitic phase transformation occurs, and the weaker martensitic interface becomes the preferred location for cracking and extension [60,61]. Similar phenomena are found in this study. Figure 14b shows that a large number of secondary cracks appear along the martensitic lath boundary in the crack propagation stage. Therefore, the beneficial effect of austenite on SCC should not be overestimated.

## 5. Conclusions

The effect of organizational evolution on the SCC of the Cr-Co-Ni-Mo series of ultra-high strength stainless steel was studied in this paper. The conclusions are as follows:(1)The precipitation of the Mo- and Cr-enriched clusters and Laves phase reduces the corrosion resistance of specimens, while the increased content of reverted austenite improves the corrosion resistance of the specimens.(2)The crack initiation of SCC for the specimens in 3.5 wt.% NaCl solution originates from pitting. The pitting is caused by the precipitation of the Mo- and Cr-enriched clusters and Laves phase during the ageing process, which results in local Mo- and Cr-depleted areas. The morphology of intergranular fractures and quasi-cleavage fractures in SCC is the result of the HE mechanism.(3)The precipitation and growth of the Mo- and Cr-enriched clusters and Laves phase lead to a decrease in the corrosion resistance and an increase in the strength of the underaged and peak-aged specimens, which then show increased SCC sensitivity. As a stable hydrogen trap in steel, austenite effectively improves the SCC resistance of the specimens. However, under the action of H and stress, the stacking fault energy of austenite decreases, martensitic phase transformation occurs, and the weaker martensitic interface becomes the preferred location for crack initiation and extension.

## Figures and Tables

**Figure 1 materials-15-00497-f001:**
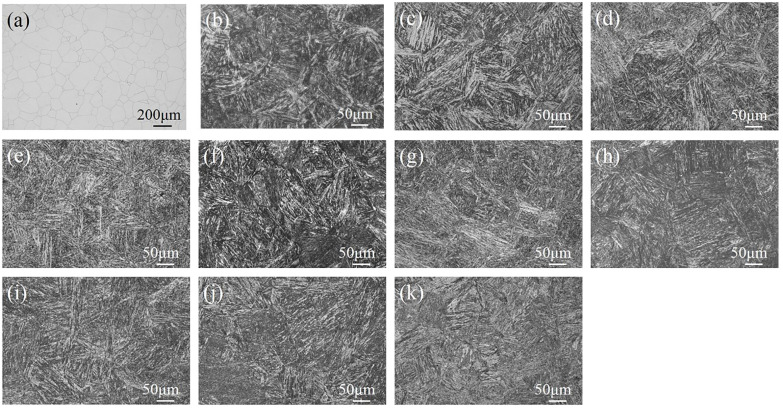
Microstructure of the specimens under various thermal conditions: (**a**) PAGBs, (**b**) DC, (**c**) 480A-0.5, (**d**) 480A-4, (**e**) 480A-80, (**f**) 540A-0.5, (**g**) 540A-4, (**h**) 540A-80, (**i**) 600A-0.5, (**j**) 600A-4, and (**k**) 600A-80.

**Figure 2 materials-15-00497-f002:**
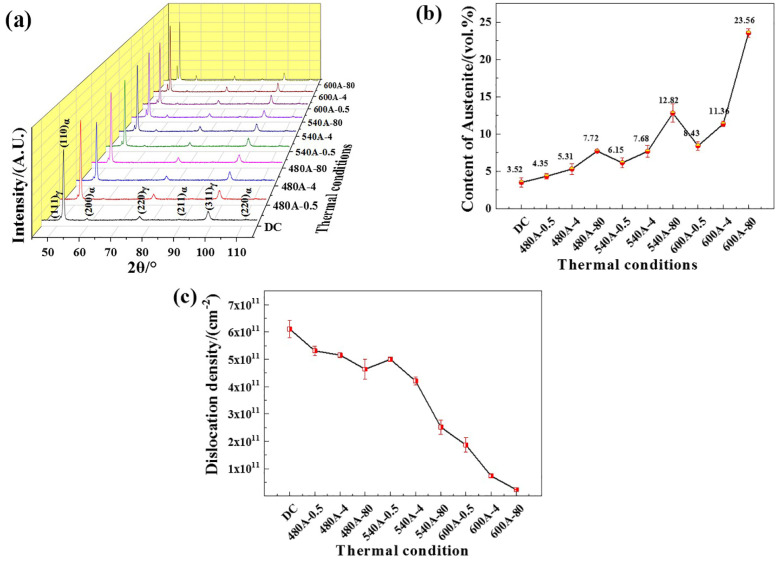
XRD analysis of the 444specimens under various thermal conditions: (**a**) XRD diffraction pattern, (**b**) volume fraction of austenite determined by XRD, and (**c**) dislocation density determined by XRD.

**Figure 3 materials-15-00497-f003:**
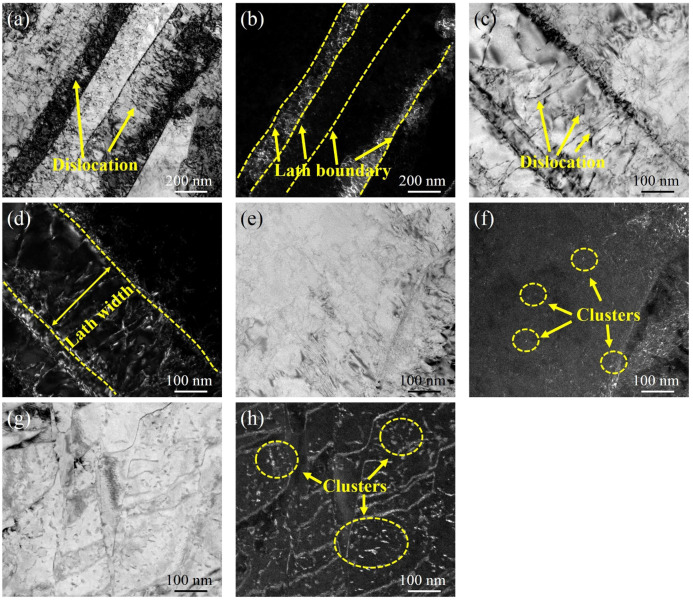
TEM morphology of the specimens under various thermal conditions: (**a**) bright-field image of DC, (**b**) dark-field image of DC, (**c**) bright-field image of 480A-0.5, (**d**) dark-field image of 480A-0.5, (**e**) bright-field image of 480A-4, (**f**) dark-field image of 480A-4, (**g**) bright-field image of 480A-80, and (**h**) dark-field image of 480A-80.

**Figure 4 materials-15-00497-f004:**
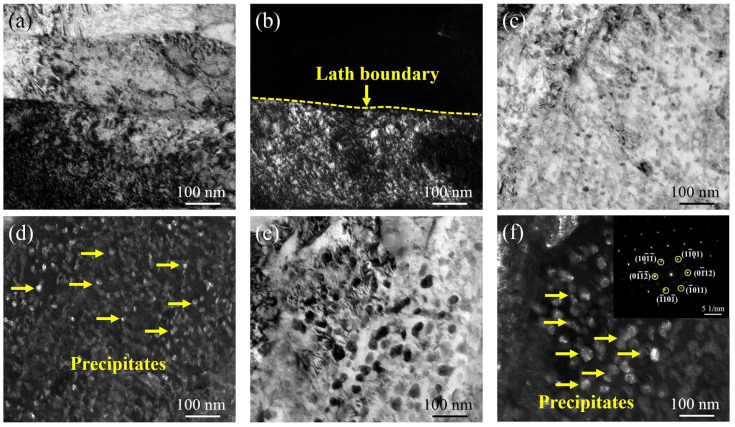
TEM morphology of the specimens under various thermal conditions: (**a**) bright-field image of 540A-0.5, (**b**) dark-field image of 540A-0.5, (**c**) bright-field image of 540A-4, (**d**) dark-field image of 540A-4, (**e**) bright-field image of 540A-80, (**f**) dark-field image of 540A-80 and the diffraction spot calibration results of the precipitates.

**Figure 5 materials-15-00497-f005:**
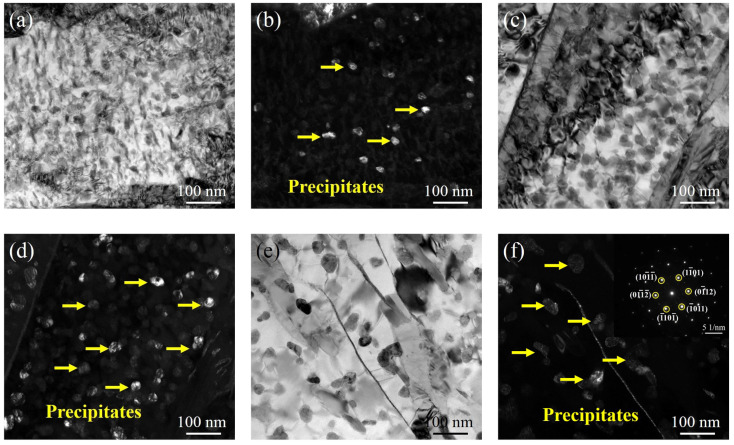
TEM morphology of the specimens under various thermal conditions: (**a**) bright-field image of 600A-0.5, (**b**) dark-field image of 600A-0.5, (**c**) bright-field image of 600A-4, (**d**) dark-field image of 600A-4, (**e**) bright-field image of 600A-80, (**f**) dark-field image of 600A-80 and the diffraction spot calibration results of the precipitates.

**Figure 6 materials-15-00497-f006:**
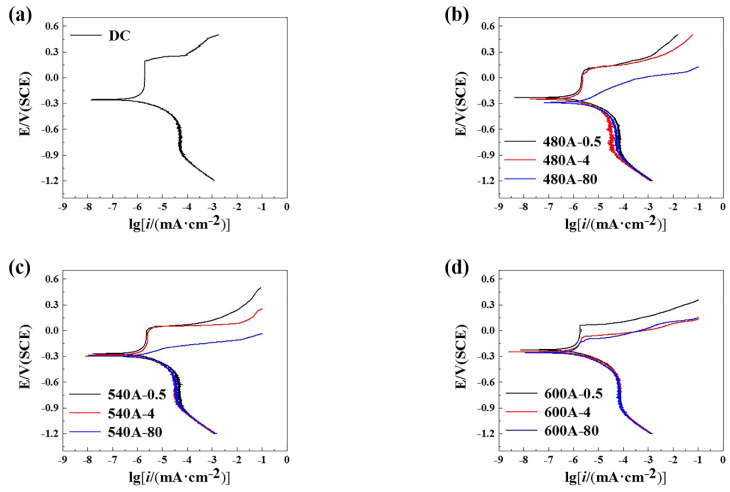
Potentiodynamic polarization curves of samples under various thermal conditions: (**a**) DC, (**b**) 480A-0.5/480A-4/480A-80, (**c**) 540A-0.5/540A-4/540A-80, and (**d**) 600A-0.5/600A-4/600A-80.

**Figure 7 materials-15-00497-f007:**
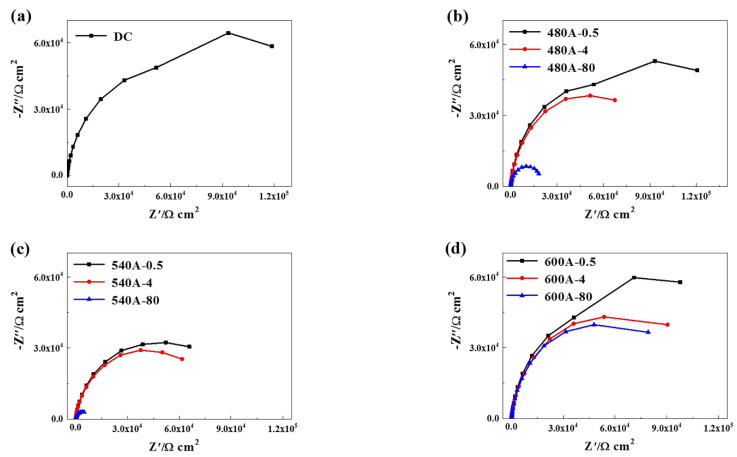
Nyquist diagram of samples under various thermal conditions: (**a**) DC, (**b**) 480A-0.5/480A-4/480A-80, (**c**) 540A-0.5/540A-4/540A-80, and (**d**) 600A-0.5/600A-4/600A-80.

**Figure 8 materials-15-00497-f008:**
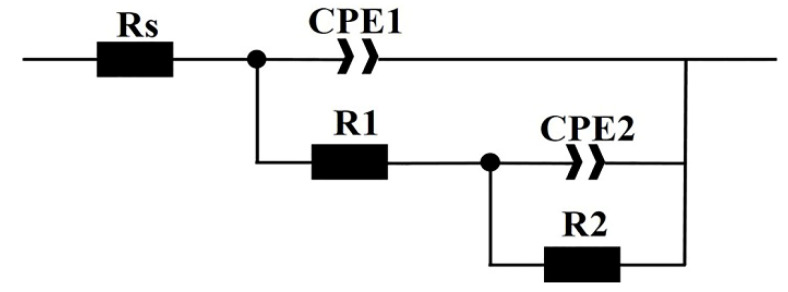
Equivalent circuit diagram.

**Figure 9 materials-15-00497-f009:**
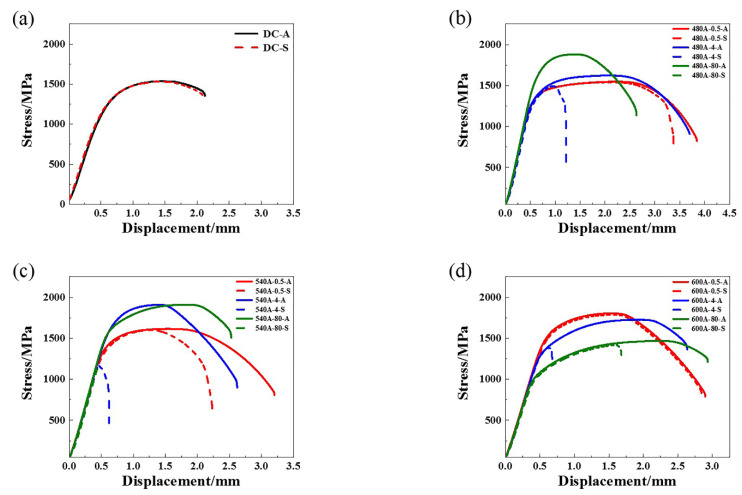
Engineering stress-displacement curves of the specimens in air and 3.5 wt.% NaCl solution: (**a**) DC, (**b**) 480A-0.5/480A-4/480A-80, (**c**) 540A-0.5/540A-4/540A-80, and (**d**) 600A-0.5/600A-4/600A-80.

**Figure 10 materials-15-00497-f010:**
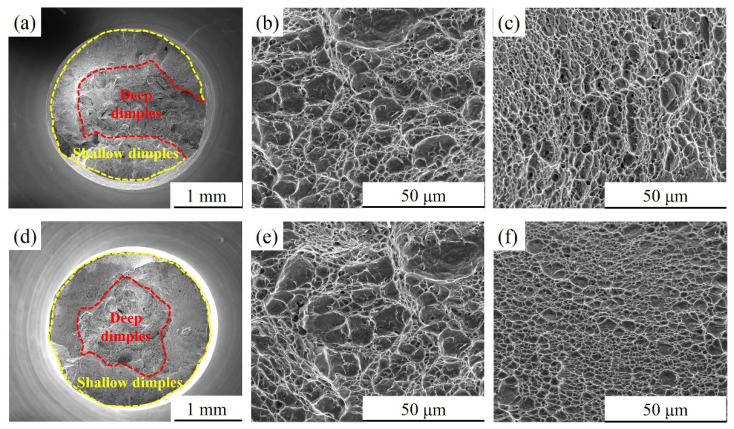
SEM fractographs of DC after SSRTs in air and 3.5 wt.% NaCl solution: (**a**–**c**) air and (**d**–**f**) 3.5 wt.% NaCl solution.

**Figure 11 materials-15-00497-f011:**
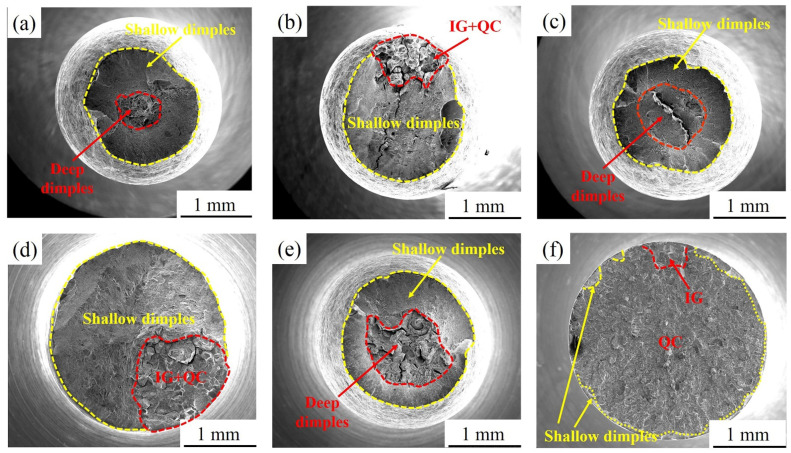
Fracture morphology of the specimens aged at 480 °C for various holding times after SSRTs in air and 3.5 wt.% NaCl solution: (**a**) air, 480A-0.5; (**b**) solution, 480A-0.5; (**c**) air, 480A-4; (**d**) solution, 480A-4; (**e**) air, 480A-80; and (**f**) solution, 480A-80.

**Figure 12 materials-15-00497-f012:**
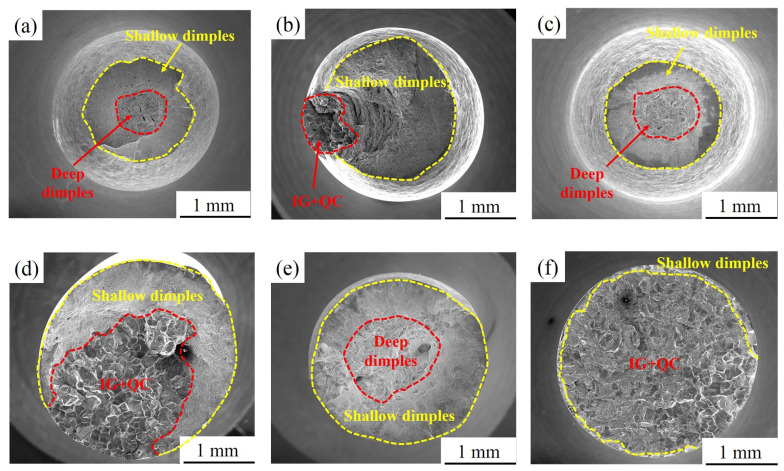
Fracture morphology of the specimens aged at 540 °C for various holding times after SSRTs in air and 3.5 wt.% NaCl solution: (**a**) air, 540A-0.5; (**b**) solution, 540A-0.5; (**c**) air, 540A-4; (**d**) solution, 540A-4; (**e**) air, 540A-80; and (**f**) solution, 540A-80.

**Figure 13 materials-15-00497-f013:**
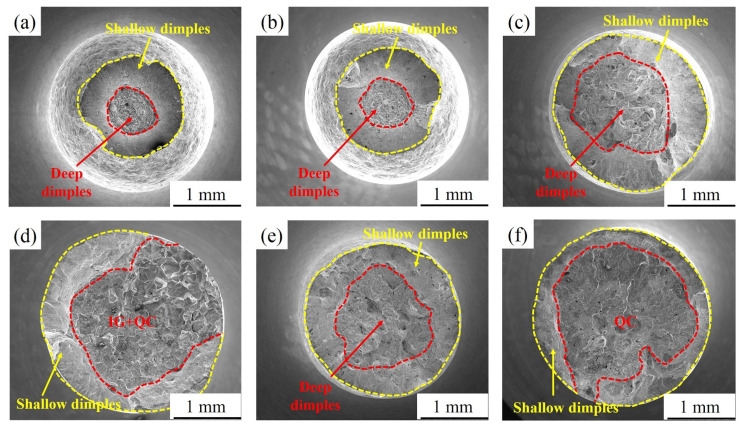
Fracture morphology of the specimens aged at 600 °C for various holding times after SSRTs in air and 3.5 wt.% NaCl solution: (**a**) air, 600A-0.5; (**b**) solution, 600A-0.5; (**c**) air, 600A-4; (**d**) solution, 600A-4; (**e**) air, 600A-80; and (**f**) solution, 600A-80.

**Figure 14 materials-15-00497-f014:**
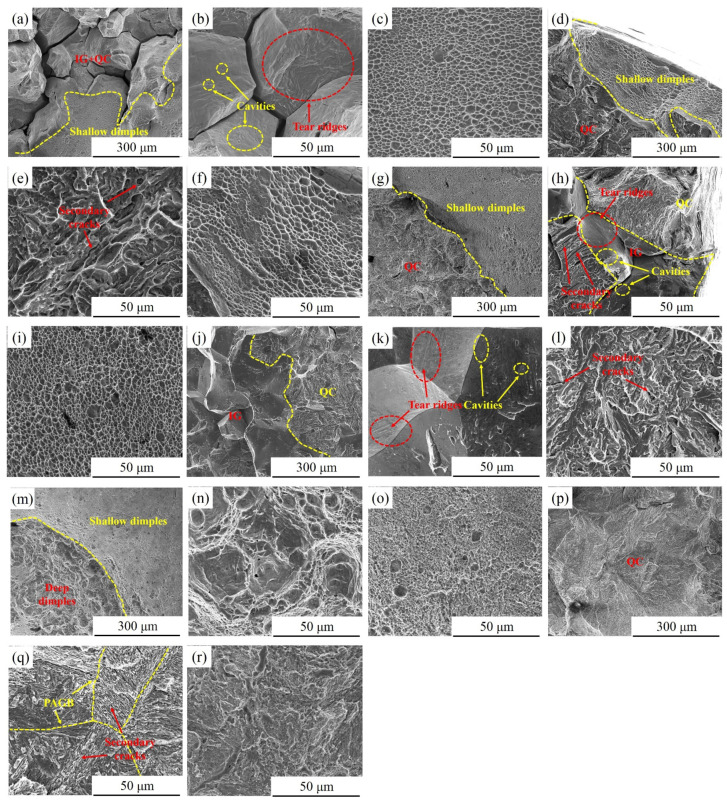
Morphology of the brittle fracture zone, plastic fracture zone and corresponding interface zone of the specimens in 3.5 wt.% NaCl solution: (**a**–**c**) 480A-0.5, (**d**–**f**) 480A-80, (**g**–**i**) 540A-0.5, (**j**–**l**) 540A-80, (**m**–**o**) 600A-0.5, and (**p**–**r**) 600A-80.

**Figure 15 materials-15-00497-f015:**
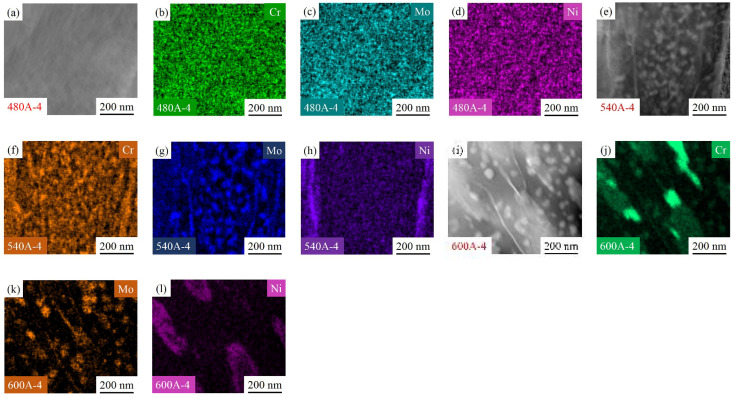
Distribution of Cr, Mo and Ni in the matrix of 480A-4 (**a**–**d**), 540A-4 (**e**–**h**) and 600A-4 (**i**–**l**).

**Figure 16 materials-15-00497-f016:**
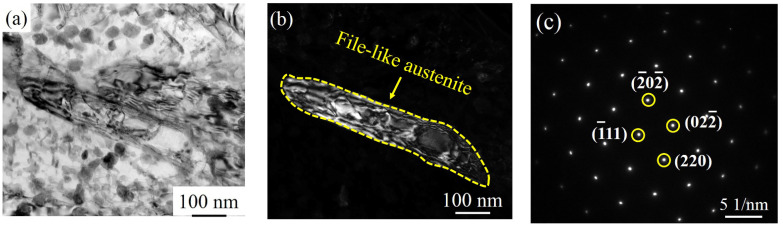
Morphology of austenite in 600A-4: (**a**) bright-field image, (**b**) dark-field image, and (**c**) diffraction spots of austenite along with the calibration results.

**Figure 17 materials-15-00497-f017:**
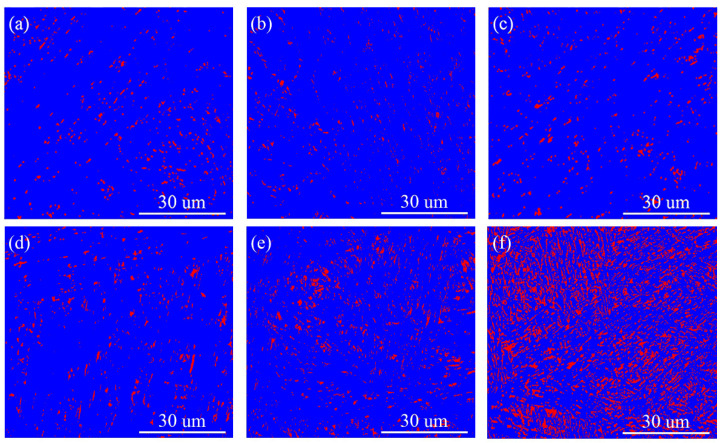
EBSD phase diagram of specimens under various thermal conditions. The blue area is the martensitic matrix, and the red area is the austenitic structure: (**a**) DC, (**b**) 480A-0.5, (**c)** 540A-0.5, (**d**) 540A-4, (**e**) 540A-80, and (**f**) 600A-80.

**Table 1 materials-15-00497-t001:** Chemical composition of the specimen.

Element	C	Cr	Ni	Mo	Co	Fe
wt.%	0.12 ± 0.003	13.5 ± 0.005	4.52 ± 0.005	5.36 ± 0.005	14.53 ± 0.005	Bal.

**Table 2 materials-15-00497-t002:** Fitting parameters of the potentiodynamic polarization curve of specimens in 3.5 wt.% NaCl solution. Icorr: corrosion current density, Ecorr: corrosion potential, IP: pitting current density, Epit: pitting potential.

Heat Treatment States	E_corr_ (V vs. SCE)	I_corr_ (uAcm^−2^)	E_pit_ (V vs. SCE)	I_p_ (µAcm^−2^)
DC	−0.255±0.02	0.125 ± 0.02	0.193 ± 0.01	1.959 ± 0.02
480A-0.5	−0.231±0.02	0.271 ± 0.02	0.087 ± 0.01	2.983 ± 0.02
480A-4	−0.249±0.02	0.326 ± 0.02	0.039 ± 0.01	3.512 ± 0.02
480A-80	−0.288±0.02	0.717 ± 0.02	-	-
540A-0.5	−0.269±0.02	0.229 ± 0.02	0.026 ± 0.01	3.016 ± 0.02
540A-4	−0.285±0.02	0.353 ± 0.02	−0.019 ± 0.01	3.787 ± 0.02
540A-80	−0.297±0.02	0.373 ± 0.02	-	-
600A-0.5	−0.223±0.02	0.171 ± 0.02	0.063 ± 0.01	1.887 ± 0.02
600A-4	−0.245±0.02	0.219 ± 0.02	−0.066 ± 0.01	2.791 ± 0.02
600A-80	−0.258±0.02	0.231 ± 0.02	−0.093 ± 0.01	4.397 ± 0.02

**Table 3 materials-15-00497-t003:** Electrochemical impedance spectrum fitting parameters of the specimens in 3.5 wt.% NaCl solution.

Heat Treatment States	Rs (Ω cm^2^)	CPE1 (Ω^−1^cm^2^s^n^)	n_1_	R1 (Ωcm^2^)	CPE2 (Ω^−1^cm^2^s^n^)	n_2_	R2 (Ωcm^2^)
DC	1.53	1.31 × 10^−5^	0.996	2.75 × 10^5^	1.56 × 10^−5^	0.823	5.61 × 10^5^
480A-0.5	1.19	2.57 × 10^−5^	0.863	1.49 × 10^5^	2.54 × 10^−5^	0.877	2.16 × 10^5^
480A-4	1.63	5.39 × 10^−5^	0.919	4.61 × 10^4^	2.87 × 10^−5^	0.878	8.38 × 10^4^
480A-80	1.45	6.88 × 10^−5^	0.915	1.55 × 10^2^	3.39 × 10^−5^	0.826	3.23 × 10^4^
540A-0.5	1.96	5.85 × 10^−5^	0.935	8.35 × 10^3^	1.27 × 10^−5^	0.829	7.99 × 10^4^
540A-4	1.85	6.14 × 10^−5^	0.926	2.05 × 10^2^	1.19 × 10^−5^	0.802	7.12 × 10^4^
540A-80	1.58	2.75 × 10^−4^	0.893	1.32 × 10^2^	1.23 × 10^−4^	0.795	6.28 × 10^3^
600A-0.5	1.22	1.92 × 10^−5^	0.837	1.62 × 10^5^	2.02 × 10^−5^	0.919	2.43 × 10^5^
600A-4	1.43	4.31 × 10^−5^	0.929	5.23 × 10^4^	2.26 × 10^−5^	0.934	1.25 × 10^5^
600A-80	1.26	4.98 × 10^−5^	0.879	3.19 × 10^4^	2.35 × 10^−5^	0.926	1.12 × 10^5^

**Table 4 materials-15-00497-t004:** Mechanical properties of the specimens under various thermal conditions in air and 3.5 wt.% NaCl solution.

Heat Treatment States	Ultimate Tensile Strength (mpa)	Elongation To Fracture (%)	Reduction of Area (%)	*δ*_loss_ (%)	*ψ*_loss_ (%)
DC-A	1536.09	7.48	22.13	0.67	0.36
DC-S	1535.53	7.43	22.05		
480A-0.5-A	1546.12	16.71	48.37	12.09	49.39
480A-0.5-S	1545.34	14.69	24.48		
480A-4-A	1621.52	16.07	46.09	67.08	83.16
480A-4-S	1494.19	5.29	7.76		
480A-80-A	1880.18	11.44	38.43	81.73	95.24
480A-80-S	1300.14	2.09	1.83		
540A-0.5-A	1615.22	13.91	45.04	19.81	31.71
540A-0.5-S	1599.29	9.73	30.76		
540A-4-A	1909.65	11.39	46.72	76.21	91.18
540A-4-S	1162.21	2.71	4.12		
540A-80-A	1908.98	10.99	19.13	82.98	99.63
540A-80-S	1121.16	1.87	0.07		
600A-0.5-A	1803.32	12.59	44.81	0.63	0.29
600A-0.5-S	1784.99	12.51	44.68		
600A-4-A	1726.82	11.46	15.55	74.26	82.06
600A-4-S	1387.97	2.95	2.79		
600A-80-A	1467.37	12.75	18.15	42.67	72.78
600A-80-S	1417.36	7.31	4.94		

**Table 5 materials-15-00497-t005:** Quantitative analysis results of the second phase in the sample.

Heat Treatment States	Mass Fraction of Elements in the Second Phase of the Specimens (wt.%)							
	Fe	Cr	Ni	Co	Mo	W	V	Σ
480A-4	0.063	0.034	0.007	0.005	0.102	0.019	0.002	0.232
540A-4	2.262	1.211	0.266	0.583	3.367	0.645	0.023	8.357
600A-4	3.591	1.923	0.419	0.916	5.361	1.021	0.038	13.269

## Data Availability

The raw/processed data required to reproduce these findings cannot be shared at this time as the data also forms part of an ongoing study.

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
