# Peer review of "Effect of Organizational Evolution on the Stress Corrosion Cracking of the Cr-Co-Ni-Mo Series of Ultra-High Strength Stainless Steel"

_materials, 2022, doi:10.3390/ma15020497_

Round 1

Reviewer 1 Report

This manuscript aims to understand the SCC susceptibility of an ultra-high strength stainless steel grade through advanced characterisation techniques and mechanical testing. Authors may address the following comments/ suggestions in the manuscript before its publication.

Page 2/ Para 2: “Li et al. [48]…” It should have been that the “resistance” to HE of the specimens correlated well with the content of austenite.

Figure 6: Please set the axis bounds and major units the same for the potentiodynamic charts. It would make visual comparison easy. Implement the same principle in other charts as well.

  1. What is organizational evolution in the current context? Please explain it in the introduction.
  2. Have you observed corrosion pits on the specimen surface? It has been suggested that pitting occurs at the Mo/ Cr depleted areas. But what are the electrochemical conditions that have setup pitting when there is no external control over the anodic or cathodic polarization? Does the straining modify the surface electrochemistry for pitting to initiate?
  3. The presence of IG and QC fracture features does suggest the act of hydrogen. Does the occluded crack chemistry at the pit bottom promote hydrogen generation and ingress?
  4. If hydrogen embrittlement is the degradation phenomenon observed in this class of stainless steel in service, then a SSRT of a H pre-charged specimen or under an in-situ H charging could have been carried out. Please explain the reason for the selection of the environment and SSRT method.
  5. Hydrogen embrittlement in stainless steels is due to the hydrogen-dislocation interactions. Do you see any link between the determined initial dislocation density and hydrogen embrittlement susceptibility?

Author Response

Response to reviewer 1

Reviewer #1: This manuscript aims to understand the SCC susceptibility of an ultra-high strength stainless steel grade through advanced characterisation techniques and mechanical testing. Authors may address the following comments/ suggestions in the manuscript before its publication.

Reply: Thank you for your careful suggestions and revisions! Specific additional contents and revisions have been shown below.

(1) Figure 6: Please set the axis bounds and major units the same for the potentiodynamic charts. It would make visual comparison easy. Implement the same principle in other charts as well.

Reply: Thank you for your valuable comment. According to your suggestion, I have modified Figures 6 and 9.

(2) What is organizational evolution in the current context? Please explain it in the introduction.

Reply: Thank you for your careful suggestions of this aspect! I have explained the organizational evolution in the current context. That is reflected in the manuscript.

(3) Have you observed corrosion pits on the specimen surface? It has been suggested that pitting occurs at the Mo/ Cr depleted areas. But what are the electrochemical conditions that have setup pitting when there is no external control over the anodic or cathodic polarization? Does the straining modify the surface electrochemistry for pitting to initiate?

Reply: Thank you for your careful suggestions and revisions! The morphology of corrosion pits on the 540A-4 specimen surface is shown in Fig.1. when there is no external control over the anodic or cathodic polarization, the nucleation of pitting is promoted by the potential difference between metal matrix and precipitation. Whether the straining modify the surface electrochemistry for pitting to initiate needs further verification. Please allow us to discuss this issue in the future study. I'm sorry for our unresolved problems.

Fig. 1 The pitting morphology of 540A-4 in 3.5 wt.% NaCl solution.

(4) The presence of IG and QC fracture features does suggest the act of hydrogen. Does the occluded crack chemistry at the pit bottom promote hydrogen generation and ingress?

Reply: Thank you for your careful suggestions of this aspect! Whether the occluded crack chemistry at the pit bottom promote hydrogen generation and ingress has been proved by Huang et al. As suggested by Huang et al. an enclosed microenvironment containing Cl-, H+, and Fe2+, among other ions, immediately develops underneath the dense product film and notably accelerates the localized dissolution process through an autocatalytic effect. Its conversion via the acid regeneration mechanism at the bottom of the corrosion pits supply large numbers of hydrogen atoms to the HSLA steel substrate [1].

(5) If hydrogen embrittlement is the degradation phenomenon observed in this class of stainless steel in service, then a SSRT of a H pre-charged specimen or under an in-situ H charging could have been carried out. Please explain the reason for the selection of the environment and SSRT method.

Reply: Thank you for your valuable comment. Ultra-high strength stainless steel is widely used in marine and other fields due to its high strength, high toughness and good corrosion resistance. The environment of 3.5 wt.% NaCl solution was chosen as the corrosion environment is to simulate the marine environment. There are two reasons for choosing the SSRT method. On the one hand, the slow strain rate method has a high sensitivity to stress corrosion cracking. On the other hand, the slow strain rate method can be used to quantitatively determine the magnitude of stress corrosion susceptibility.

(6) Hydrogen embrittlement in stainless steels is due to the hydrogen-dislocation interactions. Do you see any link between the determined initial dislocation density and hydrogen embrittlement susceptibility?

Reply: Thank you for your careful suggestions of this aspect! Hydrogen embrittlement sensitivity is related to precipitates, content of austenite and dislocation density and other comprehensive factors. In this study, no direct link between the determined initial dislocation density and hydrogen embrittlement susceptibility was found. Please allow us to discuss this issue in the future study. I'm sorry for our unresolved problems.

References

[1] Huang Y, Zhu Y. Hydrogen ion reduction in the process of iron rusting[J]. Corrosion Science, 2005, 47(6):1545-1554.

Reviewer 2 Report

This article presents comprehensive research about the impact of the thermal conditions and the respective heat treatment parameters of an ultra-high strength stainless steel, Cr-Co-Ni-Mo series, in the stress corrosion cracking. Obtained results are very relevant for the scientific community from the metallurgical point of view.

The manuscript structure is adequate and adopted bibliography is comprehensive, but in some cases is excessive or unjustified. The text should be polished to improve the English writing. Some figures required higher resolution, including the figures with plots.

It is considered that this article is in the scope of the Materials journal, but it is proposed a minor revision before its possible acceptance, aiming to clarify some parts and improve some sections.

Specific comments:

The article does not present the text numbers to assist the revision.

Page 1: The abstract is unclear about the reason or about the research question of this work. It will improve the article if it is described the problem that will be addressed.

Page 1: “Ultra-high strength stainless steel is widely used in aviation, aerospace” – please revise this sentence. Ultra-high strength stainless steel has specific applications, that can be detailed. “. In addition, aviation and aerospace is similar.

Page 2: “brought about, and the SCC susceptibility increases with increasing strength [9-14].” It is used 5 bibliographic references to support this sentence, which seems exaggerated. In these cases, it should be detailed individually the main ideas for each cited work.

Page 2: “ultra-high strength stainless steel [15-22]. “ In this case is 7 references, please revise this.

Page 3: “and forged into Φ450 mm round bars.” Please use the symbol of diameter

Page 3: “Table 1. Chemical composition of the specimen.” It is missing the measurement process and deviation in the measurements.

Page 3: “The SSRT specimens were prepared according to the requirements…” Please detail the geometry and dimensions used.

Page 3: “δA represents the elongation to fracture of the specimens in air, and δS stands for the elongation to fracture of the specimens in 3.5 wt.% NaCl solution.” How was the elongation measured?

Page 3: “Potentiodynamic polarization and electrochemical impedance spectroscopy (EIS) were performed with an electrochemical workstation” please detail the equipment used.

Page 4: “An optical micrograph of the prior austenite grain boundaries (PAGBs) of DC and the microstructure of the specimens under different thermal conditions are shown in Fig 1.” Please detail each sub-figure of Figure 1.

Page 5: Figure 2: Please improve the 2d plots. The axis numbering is not readable.

Page 10: “The fitting results of the potentiodynamic polarization curve of the specimens are shown in Table 2.” Please detail the fitting algorithm.

Page 11: Figure 9: Stress-stain curves should not be represented after break. Plot resolution should be improved.

Page 12: It is missing the Table number and the caption

Page 21: The conclusions should be improved, taking into account all the work performed.

Author Response

Response to reviewer 2

Reviewer #2: This article presents comprehensive research about the impact of the thermal conditions and the respective heat treatment parameters of an ultra-high strength stainless steel, Cr-Co-Ni-Mo series, in the stress corrosion cracking. Obtained results are very relevant for the scientific community from the metallurgical point of view.

The manuscript structure is adequate and adopted bibliography is comprehensive, but in some cases is excessive or unjustified. The text should be polished to improve the English writing. Some figures required higher resolution, including the figures with plots.

It is considered that this article is in the scope of the Materials journal, but it is proposed a minor revision before its possible acceptance, aiming to clarify some parts and improve some sections.

Specific comments:

(1) The article does not present the text numbers to assist the revision.

Reply: Thank you for your careful suggestions and revisions! According to your suggestion, I have presented the text numbers in the manuscript.

(2) Page 1: The abstract is unclear about the reason or about the research question of this work. It will improve the article if it is described the problem that will be addressed.

Reply: Thank you for your valuable comment. According to your suggestion, I have modified Figures 6 and 9. That is reflected in the manuscript.

(3) Page 1: “Ultra-high strength stainless steel is widely used in aviation, aerospace” – please revise this sentence. Ultra-high strength stainless steel has specific applications, that can be detailed. “. In addition, aviation and aerospace is similar.

Reply: Thank you for your careful suggestions of this aspect! According to your suggestion, I have detailed the specific applications of ultra-high strength stainless steel in the manuscript. That is reflected in the manuscript.

(4) Page 2: “brought about, and the SCC susceptibility increases with increasing strength [9-14].” It is used 5 bibliographic references to support this sentence, which seems exaggerated. In these cases, it should be detailed individually the main ideas for each cited work.

Reply: Thank you very much for pointing out this problem to us. Considering the reviewer's suggestion, we have deleted 3 references. That is reflected in the manuscript.

(5) Page 2: “ultra-high strength stainless steel [15-22]. “In this case is 7 references, please revise this.

Reply: Thank you for your valuable comment. we have deleted 5 references. That is reflected in the manuscript.

(6) Page 3: “and forged into Φ450 mm round bars.” Please use the symbol of diameter

Reply: Thank you for your careful suggestions of this aspect! I have checked this error and used the symbol of diameter. That is reflected in the manuscript.

(7) Page 3: “Table 1. Chemical composition of the specimen.” It is missing the measurement process and deviation in the measurements.

Reply: Thank you for your careful suggestions of this aspect! I have checked this error and complemented the measurement process and deviation in the measurements. That is reflected in the manuscript.

(8) Page 3: “The SSRT specimens were prepared according to the requirements…” Please detail the geometry and dimensions used.

Reply: Thank you for your careful suggestions of this aspect! I have detailed the geometry and dimensions of the SSRT specimens in the manuscript. That is reflected in the manuscript.

(9) Page 3: “δA represents the elongation to fracture of the specimens in air, and δS stands for the elongation to fracture of the specimens in 3.5 wt.% NaCl solution.” How was the elongation measured?

Reply: Thank you for your valuable comment. Fig. 1 shows the shape of the specimen used in the SSRTs with a diameter of 3 mm and a gauge length of 23 mm. Before the SSRTs, the length of the gauge was measured a vernier caliper for 5 times, and the average value was recorded. After the SSRTs, the length of the sample distance segment was measured again, and the average value was taken for 5 times and recorded. The δA and δS were calculated using Eq. (1)

                         (1)

Where, L0 is the length of the gauge of the sample before the test, and L1 is the length of the gauge of the sample after the test.

Fig. 1 Geometry of specimens for SSRTs

(10) Page 3: “Potentiodynamic polarization and electrochemical impedance spectroscopy (EIS) were performed with an electrochemical workstation” please detail the equipment used.

Reply: Thank you for your valuable comment. I have detailed the equipment used in our research. That is reflected in the manuscript.

(11) Page 4: “An optical micrograph of the prior austenite grain boundaries (PAGBs) of DC and the microstructure of the specimens under different thermal conditions are shown in Fig 1.” Please detail each sub-figure of Figure 1.

Reply: Thank you for your valuable comment. According to your suggestion, I have detailed each sub-figure of Figure 1. That is reflected in the manuscript.

(12) Page 5: Figure 2: Please improve the 2d plots. The axis numbering is not readable.

Reply: Thank you for your careful suggestions of this aspect! I have corrected it in the manuscript. That is reflected in the manuscript.

(13) Page 10: “The fitting results of the potentiodynamic polarization curve of the specimens are shown in Table 2.” Please detail the fitting algorithm.

Reply: Thank you for your careful suggestions of this aspect! The fitting algorithm has been described in the manuscript. That is reflected in the manuscript.

(14) Page 11: Figure 9: Stress-stain curves should not be represented after break. Plot resolution should be improved.

Reply: Thank you for your valuable comment. According to your suggestion, I have modified Figure 9. That is reflected in the manuscript.

(15) Page 12: It is missing the Table number and the caption

Reply: Thank you for your careful suggestions of this aspect! I have added the table number and the caption in the manuscript.

(16) Page 21: The conclusions should be improved, taking into account all the work performed.

Reply: Thank you very much for your sincere question. We are sorry that we have did my best to draw the conclusions. Once again, I'm sorry for our unresolved problems.
